# The INTERMAGNET framework for peer-review and activities for improvement of 1-minute Definitive Data quality and availability

Jan Reda<sup>1</sup>, Benoit Heumez<sup>2</sup>, Jürgen Matzka<sup>3</sup>

- 5 <sup>1</sup>Institute of Geophysics Polish Academy of Sciences, Warsaw, Poland
  - <sup>2</sup> Université Paris Cité- Institut de physique du globe de Paris- CNRS, Magnetic observatories, Paris, France <sup>3</sup>GFZ Helmholtz Centre for Geosciences, Potsdam, Germany

Correspondence to: Jürgen Matzka (juergen.matzka@gfz.de)

Abstract. The global INTERMAGNET network provides the scientific community with geomagnetic time series data from its more than 100 member observatories. These time series include near real-time data (Reported), as well as post-processed data products such as Quasi-Definitive and Definitive Data (hereafter referred to as QD and DD, respectively). QD data is published online with a delay of up to three months, while DD data becomes available on the INTERMAGNET website after the end of each calendar year. Additionally, the DD data and metadata are published with a unique identifier (DOI). DD data publications cover the records since 1991, and occasionally include updates of previously published DD datasets. All datasets are publicly accessible online.

DD and QD data allow the analysis of both short-term magnetic field variations and long-term secular variation. Monitoring these long-term changes distinguishes INTERMAGNET from other networks that focus primarily on short-term geomagnetic variations

INTERMAGNET ensures high-quality DD geomagnetic data through a rigorous two-stage, peer-reviewed quality control process. Although this process delays publication, it guarantees the final DD datasets' highest possible accuracy and reliability.

This article discusses key aspects of the collection, processing, and publication of DD data within the INTERMAGNET network.

## 1. Introduction




The global INTERMAGNET network (International Real-Time Magnetic Observatory Network) is a highly valuable initiative dedicated to the collection and provision of high-quality geomagnetic data (Love and Chulliat, 2013). While all INTERMAGNET observatories (IMOs) are affiliated to the International Association of Geomagnetism and Aeronomy (IAGA), not all IAGA observatories are part of INTERMAGNET. Although INTERMAGNET and IAGA serve different roles, they work closely together. IAGA establishes the scientific foundation and guidelines for observing Earth's magnetic field, whereas INTERMAGNET develops and implements the technical standards in practice, ensuring homogenous time series and the timely delivery of data. These efforts are essential for both scientific research and practical applications.

All INTERMAGNET observatories (IMOs) provide 1-minute data, while some IMOs also provide 1-second data. The data products they provide can be divided into two main categories.

- 1. **Near real-time data** these datasets, referred to as **reported data**, are transmitted almost immediately after being recorded. Their low latency offers rapid availability, however they are not yet fully calibrated or verified by experts. Near real-time data are primarily used for operational monitoring purposes, such as quality control, space weather forecasting, and the protection of critical infrastructure sensitive to high geomagnetic activity, both on Earth and in space.
- 2. Quality controlled data which include:
  - **Definitive Data** (**DD**) These are the most accurate geomagnetic records, containing both geomagnetic variations and absolute field values, corrected using absolute measurements. They are processed after the end of each calendar year. The preparation process involves the removal of artificial disturbances, and the filling of data

gaps using backup datasets when available. These datasets are thoroughly verified by experienced scientists before publication.

• Quasi-Definitive Data (QD) - Provided by INTERMAGNET observatories for over a decade, QD data are made available within three months of recording. Their quality is comparable to DD data (Peltier and Chulliat, 2010; Clarke et al., 2013). They are particularly relevant for the Swarm satellite mission (Macmillan and Olsen, 2013), as QD data are almost as accurate as Definitive Data but are available much earlier.

DD and QD data make INTERMAGNET stand out from other networks monitoring the Earth's magnetic field. They provide critical information about secular variation - slow changes in the Earth's internal magnetic field caused by core dynamics. Understanding secular variation is essential for studying the spatial and temporal properties of the geodynamo (Matzka et al., 2010). These datasets also contribute to the development of global models and maps, such as the International Geomagnetic Reference Field (IGRF) and the World Magnetic Model (WMM) (Macmillan and Quinn, 2000).

Here, we discuss the one-minute DD datasets provided by IMOs after the end of the calendar year. INTERMAGNET has established a peer-review system for these data. This cross-checking process involves expert evaluations by specialists in geomagnetic observations, ensuring that the published DD data are highly reliable and trusted by scientists, engineers, and decision-makers worldwide.

In the last section (6), we describe the policies and recent initiatives aimed at maintaining the highest data quality standards and at improving the timeliness and availability of these datasets.

#### 2. Policy for ensuring data quality






Since its inception, INTERMAGNET has emphasized the quality of data provided by observatories (Reda et al., 2011), particularly the Definitive Data set. This commitment is crucial as these data are used in scientific studies of the Earth's magnetic field and various geomagnetic models.

Only observatories that meet high-quality standards can join the INTERMAGNET network (St-Louis et al., 2020). One of the primary evaluation criteria is the analysis of baseline plots, that show the difference between the recording vector instrument and the manual absolute measurements. These plots provide important information about the quality of geomagnetic observations, including:

- Frequency and regularity of absolute measurements,
- Deviation of absolute measurements,
- Stability of the vector magnetometer used for recording geomagnetic field variations,
- Impact of external factors (e.g., seasonal, temperature) on recording equipment,
  - Method and accuracy of determining adopted baselines.

Analysis of baseline plots plays a key role in determining whether an observatory qualifies for inclusion in the INTERMAGNET network. A guide published by IAGA includes numerous references to the evaluation of baselines and their role in observatory practice (Jankowski and Sucksdorff, 1996).

Final data should meet the standards of both IAGA and INTERMAGNET. For IAGA, the most important publication about geomagnetic observatories for nearly 30 years has been the IAGA guide written by J. Jankowski and C. Sucksdorff (1996), available at: http://www.iaga-aiga.org/data/uploads/pdf/guides/iaga-guide-observatories.pdf

The specific INTERMAGNET requirements are outlined in the Technical Manual (St-Louis et al., 2020), which is regularly updated. The latest version can be found here: https://tech-man.intermagnet.org/\_/downloads/en/stable/pdf/

All Definitive Data sets are reviewed before acceptance. If any concerns arise, feedback is sent to the observatory, requesting clarification or correction. Specific data or metadata issues are addressed through direct communications with the observatory, typically via email.

A fundamental principle of INTERMAGNET's policy is that the received data is never altered – any necessary corrections must be made by the observatories themselves as part of a self-correction process. The quality control of Definitive Data is carried out by a team of volunteer experts from the Data Checking Task Team (hereafter abbreviated as DCTT), working in close collaboration with the INTERMAGNET Definitive Data Subcommittee.

#### 3. Data Checking Task Team





The DCTT is responsible for the review and verification of Definitive Data sets. This team consists of experienced volunteers, each assigned to a specific group of INTERMAGNET observatories (IMOs). The review process extends beyond analyzing the time series of geomagnetic field components, and also includes:

- Baseline values and their stability,
- Historical annual means,
- File formats and metadata.
- Compliance with INTERMAGNET standards,
- Overall data consistency.

Whenever possible, data are compared with those from nearby observatories, helping to detect unexpected jumps in absolute levels. The final approval for data publication on the INTERMAGNET website lies with the INTERMAGNET Definitive Data Subcommittee.

The review process promotes communication and knowledge sharing, benefiting all participants. INTERMAGNET's quality control system ensures that users can trust the quality and reliability of Definitive Data from the INTERMAGNET network.

More detailed information on the INTERMAGNET quality control system and the current list of DCTT members is available at: https://intermagnet.org/structure.html#data-checking-task-team. The list of IMOs and their assigned data checkers may change from year to year. A detailed list is sent to observatories at the beginning of each year as part of the "Call for Data". The "Call for Data" is an annual message sent by INTERMAGNET to observatories after the end of the calendar year. It provides the deadline for submitting Definitive Data, along with key information about data requirements, including the list of mandatory files and where to upload them. The message also highlights any changes in data standards or submission procedures compared to the previous year.

INTERMAGNET recognizes and appreciates the dedication of the DCTT members and their home institutions for their contribution to improving the quality of geomagnetic data.

# 4. Scope of control of 1-min Definitive Data set

The complete dataset submitted by observatories at the end of the year includes several files (St-Louis et al., 2020):

- Twelve final 1-minute binary data files (\*.bin), oriented XYZG where: X (north), Y (east), Z (vertical), and G is the difference between vector and scalar observations,
- One baseline file (\*.blv).
- One observatory readme file,
  - One yearmean file, listing annual mean values for the observatory,
  - One country readme file (text),
  - One About-screen (graphic) for the country file.
- These files vary in type and format, including binary files, text files and graphics files. Most are binary files, in

  INTERMAGNET Archive Format (IAF), as described in the Technical Manual (St-Louis et al., 2020), containing 1-minute data series of the XYZG geomagnetic field, and optionally K local magnetic activity index (Menvielle and Berthelier, 1991), along with essential metadata, such as:
  - IAGA code of the observatory,
  - Geographic coordinates and elevation,
- Code of the parent institution,
  - K9-limit for the local magnetic activity index (Menvielle and Berthelier, 1991),
  - Original sampling period of the recording equipment,
  - Original sensor orientation
  - Hourly means
- 130 Daily means

135

Readme text files provide information about the observatory, parent institutions, observers and responsible personnel, and observations notes. The file with the BLV extension contains valuable information about absolute measurements of the Earth's magnetic field and adopted baselines. Yearmean file contains yearly mean values observed since the beginning of the observatory's operation.

- An important aspect of quality control for geomagnetic data provided by INTERMAGNET observatories is the detection of inconsistencies within the dataset. This is particularly relevant because certain metadata, such as geographic coordinates, appear in several files of the dataset. One common issue is a discrepancy between 1-minute time series data and the annual averages contained in the yearmean file. Inconsistencies may also occur within the yearmean file itself for example, between the X (north), Y (east), and H (horizontal) components. Many other types of discrepancies are also possible.
- When reviewing 1-min data time series, the visual assessment of the magnetic field recordings is a key step to detect discrepancies between  $F(F=\sqrt{(X^2+Y^2+Z^2)})$  and S, where X, Y, and Z are recorded by the observatory's vector magnetometer, and S, which is recorded by an absolute scalar field magnetometer such as proton or Overhauser magnetometer. In addition, visual comparisons of the time series of a given observatory with those from neighboring observatories often provide valuable insights.
- Accepted Definitive Data are publicly available at:
  - 1. INTERMAGNET Website: <a href="https://imag-data.bgs.ac.uk/GIN\_V1/GINForms2">https://imag-data.bgs.ac.uk/GIN\_V1/GINForms2</a>,

    Provides 1-minute XYZF time series following approval by a DCTT volunteer and the INTERMAGNET organization.

2. DOI publication of the INTERMAGNET Reference Dataset (IRDS), available at:

https://intermagnet.org/data\_download.html#downloading\_data\_using\_dois

Dating back to 1991, it contains the most recent data updates for the entire INTERMAGNET network, along with time series, baselines, K indices, historical yearly averages and readme files. Corrections are very rare. However, occasional updates may be applied to both the one-minute average data files and the accompanying auxiliary files. Such corrections may result from, for example, errors discovered in baseline determination, mistakes in metadata (such as incorrect observatory coordinates), or the publication of incorrect annual means in yearmean-type files.

#### 5. Software for data quality control

Over several decades of INTERMAGNET's operation, various software tools have been developed for viewing, analyzing, converting formats, and verifying geomagnetic data from geophysical observatories. Some of these tools are publicly available and are regularly updated for anyone interested. A list of available software, along with download links, can be found on the INTERMAGNET website https://intermagnet.org/software.html.

160 The notable programs are:





IMCDViewer

A Java application designed to work with IAF binary files and other types of files provided by observatories at the end of the year. The program enables the visualization of minute, hourly, and daily data, baseline values, three-hour local geomagnetic activity indices (K), readme and graphical files (Dawson et al., 2009). It is useful for data quality control and allows data comparison between observatories. The program requires data to follow the structure defined for INTERMAGNET CD/DVDs. An example of data visualization using IMCDViewer is shown in Fig. 1.

MagPy

A Python-based software package for analyzing and visualizing geomagnetic data (Leonhardt et al., 2013). It allows file conversion of various formats, data plotting, and mathematical calculations related to geomagnetism, such as baseline determination and trend analysis. An example of MagPy's visualization capabilities is shown in Fig. 2.

• gm\_convert

A Java application for converting between various geomagnetic data formats: WDC, IMFV, IAGA2002, ImagCDF, and INTERMAGNET-DKA. Definitions of these formats, particularly those used by INTERMAGNET, can be found in the INTERMAGNET Technical Reference Manual (St-Louis et al., 2020). The application can operate in both graphical and command-line modes.

• check1min

A Windows console command-line tool that checks whether files comply with required formats. It focuses mainly on verifying the consistency of IAF files with other files in the yearly dataset. A key feature is the ability to compare annual averages from yearmean files with those computed from IAF files.






Fig. 1. IMCDVIEW. Data quality control by comparing the X component of two observatories (here VSS and PST) for December 2020 to January 2022.

Fig. 2. MagPy. Graphical visualization of XYZ for one month (here for Conrad Observatory WIC, DF=F-S).

## 6. Policy for improving data availability

In recent years, INTERMAGNET has made significant efforts to improve the availability and timeliness of Definitive Data. Recognising the scientific community's growing need for faster access to high-quality datasets, much effort has gone into streamlining the data review process. Volunteers from the DCTT are adopting improved workflows which lead the elaboration of condensed, standardised guidelines for data checking. They also communicate more frequently with observatories to accelerate the identification and resolution of issues. Furthermore, the process for publishing datasets with Digital Object Identifiers (DOIs) is undergoing an important evolution. Historically, INTERMAGNET would wait until datasets from all participating observatories (IMOs) were finalised before issuing the annual DOI publication. Moving forward, the DOI publication process will become incremental, allowing individual observatory datasets to be published as soon as they pass review. This change will eliminate the bottleneck caused by delays at individual observatories, ensuring that high-quality Definitive Data are made available to users more rapidly while maintaining rigorous quality standards.

## 7. Conclusion

Thanks to its rigorous quality assurance policy, two-stage verification process, and the dedication of volunteer experts, the INTERMAGNET network provides high-quality DD. The DCTT plays a key role in reviewing and validating submission, ensuring accuracy and consistency of DD. This thorough verification process makes DD a reliable source of information for both scientific research and practical applications. Although DD data undergo publication delays, they provide the accuracy and long term homogeneity of the time series that is necessary for analyzing secular changes in the Earth's magnetic field and for space climate studies. In this context, homogeneity means that variations in the data from the network of observatories should reflect true changes in the Earth's magnetic field, rather than being caused by measurement errors, equipment changes, poor calibration, or inadequate

measurement procedures at individual observatories. Further development of quality control methods and the development of analytical tools for data verification will be crucial for maintaining INTERMAGNET's high quality data standards.

#### **Author contribution**

RJ and HB. collected the information needed for the article. RJ and MJ developed the overall concept and structure of the paper. All authors contributed to editing and revising the manuscript.

## **Competing interests**





The authors declare that they have no conflict of interest.

## Acknowledgements

This work was funded by the statutory funds of Inst. Geophys. Polish. Ac. of Sc, IPGP France, and GFZ Helmholtz Centre for Geosciences.

We thank the three reviewers and the associate editor for their thoughtful comments and suggestions, which helped improve the manuscript.

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
