# Peer review of "The INTERMAGNET framework for peer-review and activities for improvement of 1-minute Definitive Data quality and availability"

_EGUsphere, 2025_

## Referee Comment (RC1)

Summary

The paper documents processes adopted by INTERMAGNET to ensure the quality and timeliness of magnetic observatory data, with emphasis on Definitive Data, I believe the key point is that a team of scientists responsible for data processing at their home observatories has been assembled who carefully check data produced by their colleagues at other observatories around the world. Besides helping to ensure data users have access to high quality data this procedure has the great benefit of spreading and building expertise among the scientists responsible for magnetic observatory data quality globally. Timeliness of publication of Definitive Data has also been improved as adoption of the peer review process imposes a timetable for review, feedback, response, agreement and data publication. Importantly the team of data checkers do not correct data; only the originators of the data make changes using feedback and advice from checkers.

Historically it has been difficult for research scientists to judge the quality of data from Individual observatories and difficult to track down the data and metadata required to make a judgement. This initiative by INTERMAGNET provides reassurance to an audience of data users about the uniform assessment procedures applied by INTERMAGNET, which is a substantial benefit for research. The allocation of DOI's to datasets published by INTERMAGNET is another significant step forward.

INTERMAGNET strives to help individual observatories work to standards that serve modern science and applications. This paper provides a good explanation of how INTERMAGNET is designing processes to improve and sustain data quality and timeliness of publication.

Comments

1. The peer review of data described in the paper is new to the magnetic observatory community and is of such significance it and might be worth emphasising by adding "by peer review" to the title?

2. Is there a better word than "delayed" (line 10). The idea is different levels of data scrutiny or quality control; very little for rea-time data; 'some' for QD and a great deal for DD. "delayed" sounds rather negative as if there's a problem, perhaps use 'quality controlled' instead

3. Definitive Data sets are intended to be final, although, very occasionally, circumstances arise where revisions are justified. It would be worth mentioning this

more explicitly, including examples of the circumstances where changes might be warranted.

4. The authors use the word "homogeneity" (Line 190). It would be helpful for them to explain clearly what is meant by this term in the context of the paper.

5. The authors abbreviate Definitive Data as DD, whereas Quasi-Definitive Data is referred to as QD. In the manuscript "DD data" is discussed that would mean Definitive Data data (e.g. Line 15). Should Quasi Definitive Data be abbreviated to QDD? Alternatively in instances where "…QD and DD data…" appear the order should be changed to "…DD and QD data … "to remove the duplication of the word 'data'.

6. The authors refer to the process of "Call for Data" (Line 99). It would be helpful for the reader to explain briefly what this process is.

Minor Comments and Corrections

1. With an acronym DD it could be better to capitalise Data in "Definitive data"?

"Definitive data" is used throughout the text and the authors should check

consistency in their use of "DD" or "Definitive data".

2. Line 33: "However, while their …"

3. Line 36: Change "Delayed Data "to "Quality controlled …" as suggested above.
It would also make sense to change the order of DD and QD bullet points since QD is available before DD.

4. Line 40: Is "magnetologist" commonly used? I might substitute "scientists".

5. Line 42: Definitive Data is referred to DD, similarly, should Quasi-Definitive Data be referred to as QDD?

Line 69: "Baseline plots analysis…" change to "Analysis of baseline plots …"

6. Line 74: "… requirements are outlined …"

7. Line 76: "…datasets…"

8. Line 86: "…components and …"

9. Line 95: "…benefiting all …"

10 Line 100: "…Data Checking Task Team members…"

11. Line 110: In "IAF format", spell out the acronym IAF.

12. Line 111: "…1-minute data series "

13. Line 111: "… K local magnetic activity index…"

14. Line 116: "K9- limit for the local magnetic activity index"

15. Line 121: "K index values"

16. Line 123: "The file with the BLV extension"

17. Line 124: "...and adopted baselines. The Yearmean file contains ..."

18. Line 129: replace "like" by "such as".

19. Line 130: Incorrect spelling of "magnetometer".

20. Line 130: "... visually comparing plots of the time ..."

21. Line 140: "...tools have..."

22. Line 141: "...converting formats..."

23. Line 156: "...INTERMAGNET-DKA"

24. Line 190. "The Data Checking Task Team (DCTT)..."

---

## Referee Comment (RC2)

**INTERMAGNET's efforts to improve the quality and availability of 1-minute Definitive Data**

Jan Reda[1], Benoit Heumez[2], Jürgen Matzka[3]

[revised manuscript text omitted]
 IAF format, containing 1-minute mean data series of the XYZF geomagnetic field or K magnetic activity índices, along with essential metadata, such as :

-    IAGA code of the observatory,
-    Geographic coordinates and elevation,
-    Code of the parent institution,
-    K9-limit for magnetic activity indices,
-    Original sampling period of the recording equipment,
-    Original sensor orientation
-    Hourly means
-    Daily means
-    K values

Readme text files provide information about the observatory, parent institutions, observers and responsible personnel, and observations notes. Files with the BLV extension contain valuable information about absolute measurements of the Earth's magnetic field and adopted baseline. Yearmean files contain yearly mean values observed since the beginning of the
observatory's operation.

Since some metadata such as geographic coordinates appears in different files, an important aspect of quality control is detecting inconsistencies within the dataset.

When reviewing 1-min data time series, the visual assessment of the magnetic field recordings is a key step to detect discrepancies between Fv ($Fv = \sqrt{Fx^2 + \phantom{Fy^2} + Fz^2}$ ) and Fs (Fs recorded by an absolute scalar magnetometer like proton or
overhauser magnetoemter). Additionally, visually comparing the time series of a given observatory with those from neighboring observatories often provides valuable insights.

Accepted Definitive data are publicly available at:

1. INTERMAGNET Website : https://imag-data.bgs.ac.uk/GIN_V1/GINForms2,

Provides 1-minute XYZF time series following approval by a DCTT volunteer and the INTERMAGNET organization.
2. DOI publication of the INTERMAGNET Reference Dataset (IRDS), available at:

https://intermagnet.org/data_download.html#downloading_data_using_dois,

It contains the most recent data updates for the entire INTERMAGNET network, dating back to 1991 along with time series, baselines, K indices, historical yearly averages and readme files.

**5. Software for data quality control**

[revised manuscript text omitted]

---

## Referee Comment (RC3)

**General Comments**

This is an informative paper discussing the methods, expectations, and value of the data products provided by participating observatories within the INTERMAGNET network. The paper recognizes the merit of the work that is done by the Data Checking Task Team while outlining the state of the standards and methods that data checkers and observatories should consider while providing data.

The scientific approach and outlined methods are valid. The paper is organized in a well-structured and balanced way that discusses related work while touching upon the scientific and societal importance of INTERMAGNET's high-quality data products.

It contains a good update and summary of the publicly available tools and software that have been made available to be able to produce and view INTERMAGNET data products.

**Specific Comments**

It may be helpful to describe more the equation for Fv or provide a citation (Line 129). I think it refers to F that is formed by X, Y, and Z from the vector magnetometer.  Following the equation for Fv, something could be said such as,
"where x, y, and z are recorded by the observatory's vector magnetometer, and Fs, which is recorded by an absolute scalar field magnetometer like a proton or Overhauser magnetometer. "

From Lines 108 and 109, I do not know if those are required or optional elements. Maybe it could be mentioned if they are optional, as well as what would be in the country read me file.

May be helpful to include  citations around Line 121 regarding K values or description of the term.

The paper has several enlightening statements such as that found in line 95 (related to the incredibly beneficial exchange of ideas) or line 39 (encouraging use of backup data when available), and lines 60-70 which explain the value of baseline plots and how they may be affected. It may be a nice place to further elaborate upon, or cite where to find, the criteria of what is considered the best, and/ or what is acceptable, when analysing the differences between baselines and data.

**Technical Corrections**

33      Suggestion for sentence rearrangement:
        "Their low latency offers rapid availability, however they are not yet fully calibrated.."

33      Suggestion to add 'purposes' after 'operational monitoring'

42      Capitalization of the word 'provided'

43      Replace the comma after the word 'recording' by a period.

74      Remove extra parenthesis in "..the Technical Manual (( St-Louis et al., 2020),"

80      Capitalization of 'data' in "Definitive data"

81      Remove 's' from "volunteers" such that it reads "volunteer experts" or switch the words
        around: "expert volunteers".

86      Remove the space between 'but also includes' and the colon. The sentence could also
        be written, "The review process extends beyond analyzing the time series of
        geomagnetic field components, and also includes:

        or

        "The review process extends beyond the analysis of the time series of
        geomagnetic field components, and also includes:"

        or

        "The review process extends beyond analyzing the time series of
        geomagnetic field components. It also includes: "

126     Remove the 's' from the word 'appears'.

130   Capitalize the 'o' in 'overhauser magntoemter' and check spelling of magnetometer.

130   Suggestion for sentence rearrangement:
        "In addition, visual comparisons of the time series of a given observatory with those from
        neighboring observatories often provide valuable insights."

136   Remove comma after link ( "..._dois,")

137   Suggestion for  sentence rearrangement:
        "Dating back to 1991, it contains the most recent data updates for the entire
        INTERMAGNET network, along with time series, baselines, K indices,..."

140   Replace 'has' with 'have' ( "...various software tools have been developed…")

143   Remove the space between the link and the comma ( "...html ,")

181  Suggestion of adding the word 'to' between 'leading' and 'the',   and remove the word 'a'
        between 'of' and  'condensed', i.e.:
        "...leading to the elaboration of condensed, standardised guidelines.."
        Or, " Volunteers from the Data Checking Task Team (DCTT) are adopting
        improved workflows which lead the elaboration of condensed, standardised guidelines.."

---

## Author Comment (AC4)

**Reply on RC1 of 18 Jul 2025**

We sincerely thank you for your thorough and insightful review of our manuscript. We truly appreciate the time and care you devoted to evaluating our work, as well as your many helpful comments, suggestions, and corrections.

Your feedback has helped us to clarify important points and improve the overall quality and precision of the manuscript. Below, we provide detailed responses to each of your specific comments, along with a description of the corresponding changes we have made.

**Referee comment 1:**

The peer review of data described in the paper is new to the magnetic observatory community and is of such significance it and might be worth emphasising by adding "by peer review" to the title?

**Reply:**

Thank you for your suggestion. We are seriously considering this suggestion. Although we have not yet decided on the final title, one option we are currently considering is: "INTERMAGNET's peer-reviewed efforts to improve the quality and availability of 1-minute Definitive Data."

**Referee comment 2:**

Is there a better word than "delayed" (line 10). The idea is different levels of data scrutiny or quality control; very little for rea-time data; 'some' for QD and a great deal for DD. "delayed" sounds rather negative as if there's a problem, perhaps use 'quality controlled' instead

**Reply:**

The word "delayed" was replaced with "post-processed," which seems like a better word in this context.

**Referee comment 3:**

Definitive Data sets are intended to be final, although, very occasionally, circumstances arise where revisions are justified. It would be worth mentioning this more explicitly, including examples of the circumstances where changes might be warranted.

**Reply:**

We have added the following clarification in Chapter 4:

Corrections are very rare. However, occasional updates may be applied to both the one-minute average data files and the accompanying auxiliary files. Such corrections may result from, for example, errors discovered in baseline determination, mistakes in metadata (such as incorrect observatory coordinates), or the publication of incorrect annual means in yearmean-type files.

Referee comment 4:

The authors use the word "homogeneity" (Line 190). It would be helpful for them to explain clearly what is meant by this term in the context of the paper.

Reply:

When using the term homogeneity, the authors refer to the consistency and uniformity of geomagnetic field measurements over time. This means that any variations in the data should reflect real changes in the Earth's magnetic field, rather than being caused by measurement errors, changes in equipment, calibration issues, or differences in procedures at individual observatories.

An explanatory sentence will be added at the point where the term homogeneity appears:

"In this context, homogeneity means that variations in the data from the network of observatories should reflect true changes in the Earth's magnetic field, rather than being caused by measurement errors, equipment changes, poor calibration, or inadequate measurement procedures at individual observatories."

Referee comment 5:

The authors abbreviate Definitive Data as DD, whereas Quasi-Definitive Data is referred to as QD. In the manuscript "DD data" is discussed that would mean Definitive Data data (e.g. Line 15). Should Quasi Definitive Data be abbreviated to QDD? Alternatively in instances where "…QD and DD data…" appear the order should be changed to "…DD and QD data … "to remove the duplication of the word 'data'.

Reply:

The abbreviation QDD suggested by the reviewer is indeed more logical. However, the abbreviation QD is the one commonly used in the geomagnetic community. Introducing a new abbreviation specifically for this article might cause confusion. That said, we truly appreciate the reviewer's suggestion and will follow their advice by changing the sequence from "DD and QD" to "QD and DD".

Referee comment 6:

The authors refer to the process of "Call for Data" (Line 99). It would be helpful for the reader to explain briefly what this process is.

Reply:

The following text will be added to Chapter 3:

The "Call for Data" is an annual message sent by INTERMAGNET to observatories after the end of the calendar year. It provides the deadline for submitting Definitive Data, along with key information about data requirements, including the list of mandatory files and where to upload them (usually the Paris GIN FTP server). The message also highlights any changes in data standards or submission procedures compared to the previous year.

Minor Comments and Corrections

1. With an acronym DD it could be better to capitalise Data in "Definitive data"?

"Definitive data" is used throughout the text and the authors should check

consistency in their use of "DD" or "Definitive data".

 Reply: has been corrected

2. Line 33: "However, while their …"

 Reply: has been corrected

3. Line 36: Change "Delayed Data "to "Quality controlled …" as suggested above. It would also make sense to change the order of DD and QD bullet points since QD is available before DD.

 Reply: has been corrected

4. Line 40: Is "magnetologist" commonly used? I might substitute "scientists".

 Reply: has been corrected

5. Line 42: Definitive Data is referred to DD, similarly, should Quasi-Definitive Data be referred to as QDD?

 Reply: The abbreviation QDD is logical, but no changes for the reasons explained earlier.

Line 69: "Baseline plots analysis…" change to "Analysis of baseline plots …"

 Reply: has been corrected

6. Line 74: "… requirements are outlined …"

 Reply: has been corrected

7. Line 76: "…datasets…"

 Reply: changed to "Definitive Data sets"

8. Line 86: "…components and …"

 Reply: has been corrected

9. Line 95: "…benefiting all …"

 Reply: has been corrected

10 Line 100: "…Data Checking Task Team members…"

 Reply: has been corrected

11. Line 110: In "IAF format", spell out the acronym IAF.

 Reply: has been corrected

12. Line 111: "…1-minute data series "

 Reply: has been corrected

13. Line 111: "… K local magnetic activity index…"

 Reply: has been corrected

14. Line 116: "K9- limit for the local magnetic activity index"

    Reply: has been corrected

15. Line 121: "K index values"

    Reply: The line was removed because K values were mentioned few lines earlier.

16. Line 123: "The file with the BLV extension"

    Reply: has been corrected

17. Line 124: "…and adopted baselines. The Yearmean file contains …"

    Reply: has been corrected

18. Line 129: replace "like" by "such as".

    Reply: has been corrected

19. Line 130: Incorrect spelling of "magnetometer".

    Reply: has been corrected

20. Line 130: "… visually comparing plots of the time …"

    Reply: has been corrected

21. Line 140: "…tools have…"

    Reply: has been corrected

22. Line 141: "…converting formats…"

    Reply: has been corrected

23. Line 156: "…INTERMAGNET-DKA"

    Reply: has been corrected

24. Line 190. "The Data Checking Task Team (DCTT)…"

    Reply: After editing the Conclusion, only the abbreviation DCTT was used.

---

## Author Response (AR1)

**Reply on RC1 of 18 Jul 2025 (anonymous)**

We sincerely thank you for your thorough and insightful review of our manuscript. We truly appreciate the time and care you devoted to evaluating our work, as well as your many helpful comments, suggestions, and corrections.

Your feedback has helped us to clarify important points and improve the overall quality and precision of the manuscript. Below, we provide detailed responses to each of your specific comments, along with a description of the corresponding changes we have made.

**Referee comment 1:**

The peer review of data described in the paper is new to the magnetic observatory community and is of such significance it and might be worth emphasising by adding "by peer review" to the title?

**Reply:**

Thank you for your suggestion. We have carefully considered this suggestion. Although we have not yet decided on the final title, one option we are currently considering is: "INTERMAGNET's peer-reviewed efforts to improve the quality and availability of 1-minute Definitive Data."

**Author's final changes:**

"The INTERMAGNET framework for peer-review and activities for improvement of 1-minute Definitive Data quality and availability"

**Referee comment 2:**

Is there a better word than "delayed" (line 10). The idea is different levels of data scrutiny or quality control; very little for real-time data; 'some' for QD and a great deal for DD. "delayed" sounds rather negative as if there's a problem, perhaps use 'quality controlled' instead

**Reply:**

The word "delayed" was replaced with "post-processed," which seems like a better word in this context.

**Author's changes:**

The relevant changes can be found in line 11.

**Referee comment 3:**

Definitive Data sets are intended to be final, although, very occasionally, circumstances arise where revisions are justified. It would be worth mentioning this more explicitly, including examples of the circumstances where changes might be warranted.

**Reply:**

We have added the following clarification in Chapter 4:

Corrections are very rare. However, occasional updates may be applied to both the one-minute average data files and the accompanying auxiliary files. Such corrections may result from, for example, errors discovered in baseline determination, mistakes in metadata (such as incorrect observatory coordinates), or the publication of incorrect annual means in yearmean-type files.

**Author's changes:**

The relevant changes can be found in lines 151-154.

**Referee comment 4:**

The authors use the word "homogeneity" (Line 190). It would be helpful for them to explain clearly what is meant by this term in the context of the paper.

**Reply:**

When using the term homogeneity, the authors refer to the consistency and uniformity of geomagnetic field measurements over time. This means that any variations in the data should reflect real changes in the Earth's magnetic field, rather than being caused by measurement errors, changes in equipment, calibration issues, or differences in procedures at individual observatories.

An explanatory sentence will be added at the point where the term homogeneity appears:

"In this context, homogeneity means that variations in the data from the network of observatories should reflect true changes in the Earth's magnetic field, rather than being caused by measurement errors, equipment changes, poor calibration, or inadequate measurement procedures at individual observatories."

**Author's changes:**

The relevant changes can be found in lines 217-219.

**Referee comment 5:**

The authors abbreviate Definitive Data as DD, whereas Quasi-Definitive Data is referred to as QD. In the manuscript "DD data" is discussed that would mean Definitive Data data (e.g. Line 15). Should Quasi Definitive Data be abbreviated to QDD? Alternatively in instances where "...QD and DD data..." appear the order should be changed to "...DD and QD data ... "to remove the duplication of the word 'data'.

**Reply:**

The abbreviation QDD suggested by the reviewer is indeed more logical. However, the abbreviation QD is the one commonly used in the geomagnetic community. Introducing a new abbreviation specifically for this article might cause confusion. That said, we truly appreciate the reviewer's suggestion and will follow their advice by changing the sequence from "QD and DD" to "DD and QD".

**Author's changes:**

The abbreviations are defined in line 12. Changing the sequence is in line 17.

**Referee comment 6:**

The authors refer to the process of "Call for Data" (Line 99). It would be helpful for the reader to explain briefly what this process is.

**Reply:**

The following text will be added to Chapter 3:

The "Call for Data" is an annual message sent by INTERMAGNET to observatories after the end of the calendar year. It provides the deadline for submitting Definitive Data, along with key information about data requirements, including the list of mandatory files and where to upload them (usually the Paris GIN FTP server). The message also highlights any changes in data standards or submission procedures compared to the previous year.

**Author's changes:**

The relevant changes can be found in lines 104-107.

**Minor Comments and Corrections**

1. With an acronym DD it could be better to capitalise Data in "Definitive data"?

"Definitive data" is used throughout the text and the authors should check consistency in their use of "DD" or "Definitive data".

Reply: has been corrected

Author's changes: checked and corrected if needed.

2. Line 33: "However, while their ..."

Reply: has been corrected

Author's changes: the relevant change in line 35

3. Line 36: Change "Delayed Data "to "Quality controlled ..." as suggested above. It would also make sense to change the order of DD and QD bullet points since QD is available before DD.

Reply: has been corrected

Author's changes: the relevant changes can be found in line 11, the word "delayed"

was replaced with "post-processed"

4. Line 40: Is "magnetologist" commonly used? I might substitute "scientists".

Reply: has been corrected

Author's changes: see line 42

5. Line 42: Definitive Data is referred to DD, similarly, should Quasi-Definitive Data be referred to as QDD?

Reply: The abbreviation QDD is logical, but no changes for the reasons explained earlier.

Line 69: "Baseline plots analysis..." change to "Analysis of baseline plots ..."

Reply: has been corrected

Author's changes: see line 72

6. Line 74: "... requirements are outlined ..."

Reply: has been corrected

Author's changes: see line 78

7. Line 76: "...datasets..."

Reply: changed to "Definitive Data sets"

Author's changes: see line 80

8. Line 86: "...components and ..."

Reply: has been corrected

Author's changes: see line 90

9. Line 95: "...benefiting all ..."

Reply: has been corrected

Author's changes: see line 99

10 Line 100: "...Data Checking Task Team members..."

Reply: has been corrected

Author's changes: see line 108

11. Line 110: In "IAF format", spell out the acronym IAF.

Reply: has been corrected

Author's changes: see line 120

12. Line 111: "...1-minute data series "

Reply: has been corrected

Author's changes: see lines 120-121

13. Line 111: "... K local magnetic activity index..."

Reply: has been corrected

Author's changes: see line 121

14. Line 116: "K9- limit for the local magnetic activity index"

Reply: has been corrected

Author's changes: see line 126

15. Line 121: "K index values"

Reply: The line was removed because K values were mentioned few lines earlier.

Author's changes: the relevant change (delete) around line 130

16. Line 123: "The file with the BLV extension" Reply: has been corrected Author's changes: see line 132 17. Line 124: "...and adopted baselines. The Yearmean file contains ..." Reply: has been corrected Author's changes: see line 133 18. Line 129: replace "like" by "such as". Reply: has been corrected Author's changes: see line 142 19. Line 130: Incorrect spelling of "magnetometer". Reply: has been corrected Author's changes: see line 142 20. Line 130: "... visually comparing plots of the time ..." Reply: has been corrected Author's changes: see line 143 21. Line 140: "...tools have..." Reply: has been corrected Author's changes: see line 156 22. Line 141: "...converting formats..." Reply: has been corrected Author's changes: see line 157 23. Line 156: "...INTERMAGNET-DKA" Reply: has been corrected Author's changes: see line 173

24. Line 190. "The Data Checking Task Team (DCTT)..."

Author's changes: see line 213

Reply: After editing the Conclusion, only the abbreviation DCTT was used.

**Reply on RC2 of 24 Jul 2025 (anonymous)**

We would like to sincerely thank you for your constructive and generally positive review of our manuscript. We greatly appreciate the time and effort you invested in providing detailed feedback, including the valuable annotations made directly in the reviewed manuscript file.

In some cases, we were not entirely sure how to interpret certain highlights or markings in the annotated file, as they were not always accompanied by specific comments. Nevertheless, wherever your intention was clear for us, we have made the appropriate revisions.

Below is a list of the changes we have made in response to your comments.

**Referee note, line 44:**

What do you mean by 'relevant'? Be more specific.

**Reply:**

The revised sentence with the word "relevant" now reads: "They are particularly relevant for the Swarm satellite mission (Macmillan and Olsen, 2013), as QD data are almost as accurate as Definitive Data but are available much earlier."

**Author's changes:**

The relevant changes can be found in lines 46-47.

**Referee note, line 98:**

The following link is much more pertinent: https://intermagnet.org/structure.html#data-checking-task-team.

Reply: has been corrected Author's changes: see line 102

**Referee note, line 100:**

Data Checking Team -> DCTT

Reply: has been corrected Author's changes: see line 108

**Referee note, line 104:**

Add descriptions to X, Y, Z and G. Particularly, G isn't familiar to non-experts of Intermagnet

**Reply:**

The revised sentence is the following: "Twelve final 1-minute binary data files (\*.bin), oriented XYZG where: X (north), Y (east), Z (vertical), and G is the difference between vector and scalar observations"

Author's changes: see lines 112-113

**Referee note, line 110:**

Add an explanation for IAF format or make a link to the relevant page of your latest Technical Manual v5.2.

**Reply:**

The revised sentence is the following: "Most are binary files, in INTERMAGNET Archive Format (IAF), as described in the Technical Manual (St-Louis et al., 2020), containing 1-minute data series of the XYZG geomagnetic field or K local magnetic activity index, along with essential metadata, such as:"

Author's changes: see lines 119-122

Referee note, lines 111, 116:

Explanation for what is K9-limit as well as K values themselves.

Reply:

A citation of the following work has been added: Menvielle, M., & Berthelier, A. (1991). The K-derived planetary indices: Description and availability. Reviews of Geophysics, 29(3), 415–432. https://doi.org/10.1029/91RG00994

Author's changes: see lines 121,126

Referee note, lines 126:

What do you mean by this sentence? If you are to say that there're still problems in Intermagnet metadata, describe them in detail.

Reply:

It would be difficult to include all possible inconsistencies in the article. However, we have expanded on the idea presented in that sentence:

An important aspect of quality control for geomagnetic data provided by INTERMAGNET observatories is the detection of inconsistencies within the dataset. This is particularly relevant because certain metadata, such as geographic coordinates, appear in several files of the dataset. One common issue is a discrepancy between 1-minute time series data and the annual averages contained in the yearmean file. Inconsistencies may also occur within the yearmean file itself - for example, between the X (north), Y (east), and H (horizontal) components. Many other types of discrepancies are also possible.

Author's changes: see lines 135-139

Referee note, line 129:

This should be a squared sum of X, Y and Z. Be consistent in your notation throughout your manuscript.

Reply:

has been corrected Author's changes: see line 141

Referee note, line 156:

Give pertinent links to IMFV and DKA in Technical Manual. If not, give explanations for each here.

Reply:

A sentence with a citation to the INTERMAGNET Technical Reference Manual was added here. This part of the text now looks as follows:

A Java application for converting between various geomagnetic data formats: WDC, IMFV, IAGA2002, ImagCDF, and INTERMAGNET-DKA. Definitions of these formats, particularly those used by INTERMAGNET, can be found in the INTERMAGNET Technical Reference Manual (St-Louis et al., 2020). The application can operate in both graphical and command-line modes.

Author's changes: see line 172

Referee note, line 173:

Add description to DF.

Reply:

Now the figure caption is as follows. Fig. 2. MagPy. Graphical visualization of XYZ for one month (here for Conrad Observatory WIC, DF=F-S).

Author's changes: see line 197

Referee note, line 185:

This is good. However, it may lead to many DOIs that rob transparency for data users. Do you have any routines that convert multiple DOIs to a single annual DOI?

Reply:

The idea is to have a single DOI. New data will continue to be added under this DOI over time. In practice, the same DOI will cover all INTERMAGNET Definitive Data sets from 1991 up to the most recently submitted and accepted data.

Author's changes: see line 207, no changes

Referee note, line 190:

Data Checking

Reply:

It was corrected by using the abbreviation DCTT that was introduced earlier.

Author's changes: see line 213

**Reply on RC3 of 18 Jul 2025 (Kristina Rossavik)**

We would like to sincerely thank you for your careful reading of our manuscript and for your thoughtful and constructive comments. We are pleased to know that you found the scientific approach and presentation satisfactory.

Below, we address each of your specific suggestions aimed at improving the clarity and completeness of the paper. We appreciate your input and have made revisions accordingly.

**SPECIFIC COMMENTS**

**Referee comment 1:**

It may be helpful to describe more the equation for Fv or provide a citation (Line 129). I think it refers to F that is formed by X, Y, and Z from the vector magnetometer. Following the equation for Fv, something could be said such as, "where x, y, and z are recorded by the observatory's vector magnetometer, and Fs, which is recorded by an absolute scalar field magnetometer like a proton or Overhauser magnetometer."

**Reply:**

Thank you for the suggestion. The text around line 129 now reads as follows. When reviewing 1-min data time series, the visual assessment of the magnetic field recordings is a key step to detect discrepancies between F (F=sqrt( $X^2+Y^2+Z^2$ ) and S, where X, Y, and Z are recorded by the observatory's vector magnetometer, and S, which is recorded by an absolute scalar field magnetometer like a proton or Overhauser magnetometer.

Author's changes: see lines 140-144

**Referee comment 2:**

From Lines 108 and 109, I do not know if those are required or optional elements. Maybe it could be mentioned if they are optional, as well as what would be in the country read me file.

**Reply:**

It's true that one might wonder whether these files are required or simply recommended. INTERMAGNET continues to collect them from observatories, which is relatively easy when all observatories in a given country are managed by the same parent institution. In this article, however, we prefer not to go into this topic in detail.

Author's changes: see lines 117-118, no changes

**Referee comment 3:**

May be helpful to include citations around Line 121 regarding K values or description of the term.

**Reply:**

A citation has been added to the local K index and the K9-limit:

Menvielle, M., & Berthelier, A. (1991). The K-derived planetary indices: Description and availability. Reviews of Geophysics, 29(3), 415–432. https://doi.org/10.1029/91RG00994

Author's changes: see lines 121, 126

**Referee comment 4:**

The paper has several enlightening statements such as that found in line 95 (related to the incredibly beneficial exchange of ideas) or line 39 (encouraging use of backup data when available), and lines 60-70 which explain the value of baseline plots and how they may be affected. It may be a nice place to further elaborate upon, or cite where to find, the criteria of what is considered the best, and/ or what is acceptable, when analysing the differences between baselines and data.

**Reply:**

Thank you for your kind comments. As for the suggestion to expand on the topic of baselines, this is indeed quite a broad subject, and it would be difficult to cover it in more detail within the scope of this article. We believe it would be better suited for a separate publication. For that reason, we have added a sentence with a reference to the IAGA Guide (Jankowski, J., and C. Sucksdorff, 1996, Guide for Magnetic Measurements and Observatory Practice, Int. Assoc. of Geomagn. and Aeron., Warsaw), where this topic is addressed in several places. The added sentence reads as follows:

"A guide published by IAGA includes numerous references to the evaluation of baselines and their role in observatory practice (Jankowski and Sucksdorff, 1996)."

Author's changes: see lines 73-74

**TECHNICAL CORRECTIONS**

Line 33: Suggestion for sentence rearrangement: "Their low latency offers rapid availability, however they are not yet fully calibrated.."

Reply: has been corrected

Author's changes: see line 35

Line 33: Suggestion to add 'purposes' after 'operational monitoring'

Reply: has been corrected

Author's changes: see line 36

Line 42: Capitalization of the word 'provided'

Reply: has been corrected

Author's changes: see line 44

Line 43: Replace the comma after the word 'recording' by a period.

Reply: has been corrected

Author's changes: see line 45

Line 74: Remove extra parenthesis in "..the Technical Manual ((St-Louis et al., 2020),"

Reply: has been corrected

Author's changes: see line 78

Line 80: Capitalization of 'data' in "Definitive data"

Reply: has been corrected

Author's changes: see line 84

Line 81: Remove 's' from "volunteers" such that it reads "volunteer experts" or switch the

words around: "expert volunteers".

Reply: has been corrected

Author's changes: see line 85

Line 86: Remove the space between 'but also includes' and the colon. The sentence could also be written.

"The review process extends beyond analyzing the time series of geomagnetic field components, and also includes:

or

"The review process extends beyond the analysis of the time series of geomagnetic field components, and also includes:"

or

"The review process extends beyond analyzing the time series of geomagnetic field components. It also includes:"

Reply: We followed the first suggestion.

Author's changes: see lines 89-90

Line 126: Remove the 's' from the word 'appears'.

Reply: has been corrected

Author's changes: see line 136

Line 130: Capitalize the 'o' in 'overhauser magntoemter' and check spelling of magnetometer.

Reply: has been corrected

Author's changes: see lines 141, 142

Line 130: Suggestion for sentence rearrangement: "In addition, visual comparisons of the time series of a given observatory with those from neighboring observatories often provide valuable insights."

Reply: We used the suggestion. Thank you.

Author's changes: see lines 142-144

Line 136: Remove comma after link ("... dois,")

Reply: has been corrected

Author's changes: see line 149

Line 137: Suggestion for sentence rearrangement: "Dating back to 1991, it contains the most recent data updates for the entire INTERMAGNET network, along with time series, baselines, K indices,..."

Reply: We used the suggestion. Thank you.

Author's changes: see lines 150-151

Line 140: Replace 'has' with 'have' ("...various software tools have been developed...")

Reply: has been corrected

Author's changes: see line 156

Line 143: Remove the space between the link and the comma ("...html,")

Reply: has been corrected

Author's changes: see line 159

Line 181: Suggestion of adding the word 'to' between 'leading' and 'the', and remove the word 'a' between 'of' and 'condensed', i.e.: "...leading to the elaboration of condensed, standardised guidelines.." Or, "Volunteers from the Data Checking Task Team (DCTT) are adopting improved workflows which lead the elaboration of condensed, standardised guidelines.."

Reply: We used the suggestion (option 2). Thank you.

Author's changes: see lines 203-204

**Associate Editor remarks of June 20, 2025**

The article presents the INTERMAGNET peer-reviewed geomagnetic observatory definitive data process. An important process that ensures high quality data for the geomagnetic community. The article fits very well to the special issue "Geomagnetic observatories, their data, and the application of their data", and is well structured and presented. It is ready for review and discussion. However, I have a few suggestions for very minor technical corrections that the authors may wish to consider.

Line 55: add "." in the end of the sentence

Reply: Done

Author's changes: see line 58

Line 84, 180 and 190: DCTT already defined in line 81 (revised version)

Reply: DCTT introduced 1-st time in line 85,

Author's changes: see lines 85, 88, 101, 108, 147, 203, 213

Line 129: Fx, Fy and Fz is introduced here, I suggest replacing with X, Y and Z for consistency

Reply: Now X, Y, Z, S is introduced

Author's changes: see lines 141-142

Line 190: DD already defined on line 37

Reply: Now DD and QD is defined

Author's changes: see line 12

Line 197: Please only use initials instead of full author names in this section

Reply: Done

Author's changes: see lines 222-223